# Direct Targeting *KRAS* Mutation in Non-Small Cell Lung Cancer: Focus on Resistance

**DOI:** 10.3390/cancers14051321

**Published:** 2022-03-04

**Authors:** Damien Reita, Lucile Pabst, Erwan Pencreach, Eric Guérin, Laurent Dano, Valérie Rimelen, Anne-Claire Voegeli, Laurent Vallat, Céline Mascaux, Michèle Beau-Faller

**Affiliations:** 1Department of Biochemistry and Molecular Biology, Strasbourg University Hospital, CEDEX, 67098 Strasbourg, France; damien.reita@chru-strasbourg.fr (D.R.); erwan.pencreach@chru-strasbourg.fr (E.P.); eric.guerin@chru-strasbourg.fr (E.G.); laurent.dano@chru-strasbourg.fr (L.D.); valerie.rimelen@chru-strasbourg.fr (V.R.); anne-claire.voegeli@chru-strasbourg.fr (A.-C.V.); laurent.vallat@chru-strasbourg.fr (L.V.); 2Bio-Imagery and Pathology (LBP), UMR CNRS 7021, Strasbourg University, 67400 Illkirch-Graffenstaden, France; 3Department of Pneumology, Strasbourg University Hospital, CEDEX, 67091 Strasbourg, France; lucile.pabst@chru-strasbourg.fr (L.P.); celine.mascaux@chru-strasbourg.fr (C.M.); 4Laboratory Streinth (STress REsponse and INnovative THerapy Against Cancer), Université de Strasbourg, Inserm UMR_S 1113, IRFAC, ITI InnoVec, 3 Avenue Molière, 67200 Strasbourg, France

**Keywords:** *KRAS* mutations, non-small cell lung cancer, *KRAS* G12C inhibitors, resistance mechanisms, phenotypic changes

## Abstract

**Simple Summary:**

*KRAS* is the most frequently mutated oncogene in non-small cell lung cancers (NSCLC), with a frequency around 30%, and among them KRAS G12C mutation occurs in 11% of cases. *KRAS* mutations were for a long time considered to be non-targetable alterations or “undruggable”. Direct inhibition is actually developped with switch-II mutant selective covalent KRAS G12C inhibitors with small molecules such as sotorasib or adagrasib preventing conversion of the mutant protein to GTP-bound active state. Little is known about primary or acquired resistance. Acquired resistance does occur and could be related to genetic alterations in the nucleotide exchange function or adaptive mechanisms either in down-stream pathways or in newly expressed *KRAS* G12C mutation. Mechanisms of resistance could be classified as “on-target” mechanisms, involving KRAS G12C alterations, or “off-target” mechanisms, involving other gene alterations and/or phenotypic changes.

**Abstract:**

KRAS is the most frequently mutated oncogene in non-small cell lung cancers (NSCLC), with a frequency of around 30%, and encoding a GTPAse that cycles between active form (GTP-bound) to inactive form (GDP-bound). The *KRAS* mutations favor the active form with inhibition of GTPAse activity. *KRAS* mutations are often with poor response of EGFR targeted therapies. *KRAS* mutations are good predictive factor for immunotherapy. The lack of success with direct targeting of KRAS proteins, downstream inhibition of KRAS effector pathways, and other strategies contributed to a focus on developing mutation-specific KRAS inhibitors. *KRAS* p.G12C mutation is one of the most frequent KRAS mutation in NSCLC, especially in current and former smokers (over 40%), which occurs among approximately 12–14% of NSCLC tumors. The mutated cysteine resides next to a pocket (P2) of the switch II region, and P2 is present only in the inactive GDP-bound KRAS. Small molecules such as sotorasib are now the first targeted drugs for *KRAS* G12C mutation, preventing conversion of the mutant protein to GTP-bound active state. Little is known about primary or acquired resistance. Acquired resistance does occur and may be due to genetic alterations in the nucleotide exchange function or adaptative mechanisms in either downstream pathways or in newly expressed *KRAS* G12C mutation.

## 1. Introduction

Lung cancer remains the leading malignancy worldwide in terms of incidence and mortality. NSCLC represents 85% of lung cancers and is divided into adenocarcinoma (ADC), 40%; squamous-cell carcinoma (SCC), 30%; and large cell carcinoma (LCC), 10% [1,2]. The best improvement of prognosis in NSCLC is based on new treatments as targeted therapies and immune check-points inhibitors [3,4]. The oncogenic driver is defined to play a crucial role in the disease biological pathway. Oncogenic addiction of a tumor drives a dramatic response to a targeted therapy [5]. New precision medicine is based on molecular characteristics of tumors with the detection of biomarkers of oncogenic drivers having predictive value. Such medicine is based on the existence of predictive molecular alterations, diagnostic assays, and effective targeted therapeutic agents. Molecular diagnosis of lung ADC is essential to detect such biomarkers [6].

In lung ADC, the first example of successful targeted therapy is the pharmacological inhibition of the epidermal growth factor receptor (EGFR) with increasing survival of patients with actionable tumor *EGFR* mutations receiving EGFR tyrosine kinase inhibitors (TKI) [7]. The accelerated approval of the first anaplastic lymphoma kinase (ALK) TKI crizotinib increases the paradigm of cancer drugs development. New molecular techniques applied for biomarker detection in lung tumors, such as mutations or gene fusions, favored the development of new effective drugs targeting other oncogenic drivers, such as *ROS-1* fusions, *BRAF* mutation V600E, and other rare alterations such as *MET* delta 14 mutations, *RET* fusions, *NTRK* fusions.

Tumor genotyping has now been incorporated in the clinical management of NSCLC with routine testing for *EGFR*, *KRAS*, *BRAF*, *HER2*, *ALK*, *ROS-1*, and *RET* to personalize first or second-line treatment [8]. Other molecular analysis is a part of boarding testing panels to identify some other rare actionable alterations to be targeted.

Among molecular alterations of lung ADC, *KRAS* mutations are a very frequent alteration, representing up to one third of molecular abnormalities. Despite this high frequency, *KRAS* mutations have a very limited and uncertain role as prognostic or predictive biomarkers in patients with NSCLC, and precision medicine does not appear to be an adequate approach for this subgroup of patients. *KRAS* mutations were for a long time considered non-targetable alterations or “undruggable”. After preclinical studies, results of early clinical trials recently demonstrated that pharmacological inhibition of KRAS G12C mutated protein is feasible, opening the possibility of a new targeted treatment for large subset of patients with advanced NSCLC [9,10,11,12,13,14,15]. In this review, we report the biological basis and prevalence of *KRAS* mutations in lung cancer, the molecular techniques for identification of such mutations, the overview of actual therapeutic strategies, the primary and secondary resistance to targeted *KRAS* agents, and the perspective of KRAS inhibition.

### 1.1. KRAS Mutations in NSCLC

*RAS* (*KRAS*, *NRAS*, and *HRAS*) represents the most frequently mutated gene family in human cancers [16]. Among them, *KRAS* is one of the most isoform, present in 85% of pancreatic ADC, 45% of colo-rectal, and 30% of lung ADC, and consequently is a common oncogene driver [17].

### 1.2. KRAS Signaling Pathway

The *KRAS* gene is located on chromosome 12p11.1-11.2 and consists of six exons (Figure 1A). It encodes a 21 KDa little monomeric protein G with little GTPase intrinsic activity. This protein has three major domains; one is the G-domain, containing switch I and switch II loops, a highly conserved domain responsible for GDP-GTP exchange. In its active state, KRAS transmits signals from the cell membrane to the nucleus, activating numerous signaling pathways following receptor tyrosine kinase (RTK) activation, (EGFR, ALK, or cMET …) and finally leading to the activation of transcription factors which lead to the regulation of cell growth (cell proliferation and cell survival) and differentiation (Figure 2A).

The KRAS protein normally functions as a molecular switch (Figure 2B). The inactivation and activation states are determined by the nucleoside guanosine guanosine-diphosphate (GDP) or by the nucleoside guanosine triphosphate (GTP) binding, respectively. In normal cells, KRAS is a small switch signaling GTPase between GTP-bound active state and GDP-bound inactive state. KRAS cellular signaling state depends on their activation by guanine exchange factors (GEFs) complexed with SOS (son of sevenless isoforms, SOS1 protein)—GRB2 (Growth factor Receptor Bound protein-2)—SHP2 (Src homology phosphatase 2) which catalyze the loading of GTP. Deactivation is facilitated by GTPase activating proteins (GAPs as p120GAP, neuro-fibromin NF1, or others) that increase intrinsic GTP hydrolysis of KRAS (Figure 2A,B). Binding of EGF to EGFR induces dimerization of the receptor, followed by auto-phosphorylation and trans-phosphorylation. The phosphorylated receptor binds to the adaptor protein GRB2. This complex is located near and recruits SOS proteins to the plasma membrane. Once recruited to the plasma membrane, SOS is capable of displacing GDP from RAS, allowing RAS–GTP interaction. RAS can also regulate SOS activity, suggesting that the pathway is bi-directional. Protein tyrosine phosphatase SHP2 activates KRAS downstream of RTKs through several mechanisms, including de-phosphorylation of Sprouty proteins. *NF1* mutations could be germline and predisposed to a variety of tumors, and more recently somatic in various cancers such as lung adenocarcinoma [18]. The binding of GTP to RAS induces changes in switch I and switch II loops of the G-domain, thereby activating RAS. 

In its activating state, KRAS-GTP directly interacts and activates several downstream effector proteins such as RAF and PI3K (Figure 2A). RAF is a serine threonine kinase with three sub-types, A-RAF, B-RAF, and C-RAF. Binding GTP to RAS promotes the recruitment of RAF to the cell membrane, dimerization of RAF, and phosphorylation. Activated RAF phosphorylates MEK or Mitogen-Activated Protein Kinase (MAPK), which phosphorylates Extra-cellular signal Regulated Kinase (ERK), driving cell progression and proliferation. B-RAF is frequently mutated in human cancers as lung adenocarcinoma, has a higher basal kinase activity and is easily activated by RAS. The second best characterized RAS effector is phosphatidylinositol 3-kinase (PI3K), which ultimately activates mTOR. mTOR activates the translation factor S6K driving regulation of apoptosis, metabolism, and translation. 

For reversible implementation of this pathway, RAS proteins oscillate between an active GTP-bound and an inactive GDP-bound state at rates controlled by up-stream growth-factor-dependent signals. The KRAS protein cycles between GTP and GDP-bound states with a resynthesis half-life of 24 h [19]. 

### 1.3. Molecular Epidemiology of Activating KRAS Mutations in NSCLC

*KRAS* mutations result in a single amino-acid substitution that activate RAS protein by limiting its ability to hydrolyze GTP [20]. These mutations disrupt the guanine exchange cycle, with loss of GTPase activity and GAPs dependency leading to a KRAS GTP-bound active state. Activating KRAS mutations results in the accumulation of active GTP-bound RAS able to activate several downstream signaling pathways with tumor cell proliferation. There is a high MAPK pathway activation. The receptor tyrosine kinase (RTK) activation is suppressed by ERK-mediated-negative feedback. 

Mutations in exons 2, 3, and 4 of *KRAS* lead to such a constitutive activation of KRAS, independently of activation of upstream proteins. About 90% of *KRAS* mutations are detected in codon 12 (exon 2), and even other KRAS mutations are located in codon 13, 61, 117, and 146 (exon 2 and 3) all around the nucleotide-binding pocket (Figure 1A). Substitution at the G12 (D/C/V/R/A/S) or G13 (D/C/V) residues prevent stabilization of the hydrolysis transition state. Less common variants such as Q61H/L/R/K interfere with coordination of the molecule involved in hydrolysis, whereas A146T enhances the propensity for nucleotide exchange (Figure 2B) [20].

The molecular epidemiology of *KRAS* mutation is well known in NSCLC. *KRAS* mutations are more common in ADC (20–40%) and less common in SCC (<5%) [21]. *KRAS* mutations are more common in smokers versus non-smokers (30% versus 11%) and in Western versus Asian patients (26% versus 11%).

The most frequent somatic *KRAS* mutation in NSCLC is *KRAS* G12C, which is also common in colo-rectal cancer (KRAS G12C results from nucleotide transversion with a guanine (G) replaced by a thymine (T) at coding position 34 GGT->TGT (c.34G>T) leading to replacement of amino acid glycine (G) to cysteine (C) (p.Gly12Cys). Although insensitive to GAP-assisted hydrolysis, KRAS-mutated oncoproteins have measurable intrinsic GTP hydrolysis rates, could cycle between their active and inactive states in cancer cells, and are dependent on nucleotide exchange for activation [20].

*KRAS* G12C mutations represent 39–42% of *KRAS* mutations (followed by G12V mutation) and 11–13% of total molecular alterations observed in NSCLC [22]. KRAS G12C are more frequent in smokers versus non-smokers due to tobacco carcinogens such as KRAS G12D (transition from G to adenine (A), GGT>GAT, c.35 G>A; amino acid G to asparatic acid (D), p.Gly12Asp) [23] (Figure 1B). Distribution of *KRAS* G12C somatic mutations in NSCLC could also vary among race and sex. White and black patient groups are enriched for KRAS G12C mutations more than Asians, and more often in white female patients than in white male patients, and more often in Asian male than in Asian female patients [24]. There is no evidence based data to explain these differences. 

### 1.4. Biochemical Property of KRAS G12C Mutation

Intrinsic GTPase and GDP–GTP exchange rates can vary among the different *KRAS* mutants [13]. KRAS G12C is located on the P-loop and is implicated in the nucleotide stabilization during the activation step, leading to the alteration of both intrinsic and GAP-inducing hydrolysis and not altering the rate of nucleotide exchange [25] (Figure 2B). More precisely, KRAS G12C exhibits near wild-type intrinsic GTPase activity despite its reduced p120 GAP-mediated hydrolysis state. This biochemical property of KRAS G12C is used to target KRAS-G12C by covalent inhibitors that bind to the GDP-bound state of KRAS-G12C [13]. By contrast, KRAS Q61 has elevated intrinsic exchange activity, suggesting that the GDP-bound state is short-lived. KRAS mutation in codon 13 is partially sensitive to NF1 GAP hydrolysis, whereas KRAS mutations in codon 12 are insensitive to NF1 [26]. Finally, these biochemical differences between KRAS mutations sub-types will determine which nucleotide-bound state of KRAS would therefore be most appropriate to target with allele-specific inhibitors. Low levels of GTPase activity or high levels of GEF could pose difficulties targeting the GDP-bound state. 

The mutant KRAS G12C protein, although mostly engaged in its active conformation, still undergoes nucleotide cycling and experiences periods of inactivity, which allows KRAS G12C inhibitor drugs trapping and covalent attacks [13].

### 1.5. KRAS Co-Mutations

*KRAS* mutated NSCLC represent a heterogeneous genetic group with different pattern of co-mutations [17]. These co-mutations appear more frequently in smokers. Co-occurring genomic alterations in *KRAS*-mutated tumors have an effect on the tumor biology and response to systemic therapies [27].

About half of *KRAS* mutations harbored additional concomitant mutations in different oncogene, with *TP53* (39–42%), *STK11* (20–29%), *KEAP1* (13–27%), *ATM* (13%), *cMET* 15.4%), *ERBB2* (13.8%), and *CDKN2A/B* among those commonly reported [27,28]. *STK11* and *KEAP1* are associated with inferior treatment outcomes and a poor prognosis in NSCLC patients [17,27,28,29].

These co-mutations could act via intrinsic RAS signaling pathways as well as the immune infiltration of the tumors. For example, *KRAS/TP53* co-mutated tumors are associated with high CD8+ T-cells infiltration density (defined as “hot” tumors) [30]. By contrast, *KRAS/STK11/KEAP1* co-mutated tumors are associated with reduced density of infiltrating cytotoxic CD8^+^ T lymphocytes microenvironment (defined as “cold” tumors), less likely to respond to immunotherapy [27]. Interestingly, as demonstrated by Ricciuti et al., STK11 and KEAP1 mutations seem to confer worse outcomes to immunotherapy only among patients with KRAS^MUT^, not among KRAS^WT^ [31]. 

*KRAS* mutation are typically considered as mutually exclusive with other molecular driver alterations in NSCLC as *EGFR* mutations, or *ALK*, *ROS1* fusions. 

### 1.6. KRAS Mutations and Microenvironment

Besides tumor growth, KRAS could play a role in interactions between tumor cells and the microenvironment, which could affect therapeutic response. *KRAS* mutated cell lines have been associated with decreased major histocompatibility class I expression (MCH I), upregulation of programmed death ligand 1 expression (PD-L1), and promotion of an immunosuppressive immune cell population as myeloid-derived suppressor cells (MDSCs) [31,32].

### 1.7. Clinical Value of KRAS Mutations in NSCLC

Prognostic value of *KRAS* mutations are controversial [32,33]. *KRAS* mutated NSCLC patients seem to be a bad prognosis factor in NSCLC as related by meta-analysis [33,34]. Nevertheless, prognostic value appears different depending on *KRAS* mutations. Patients with KRAS G12C mutation seem to have longer survival rates than those with other types of KRAS mutations [35].

If a *KRAS* mutation appears to be a negative predictive factor to targeted therapies such as EGFR-TKI, ALK-TKI, or MET-TKI, these results are controversial, perhaps due to different frequencies of *KRAS* mutations [36,37]. No predictive value of *KRAS* mutation has been reported for VEGF inhibitors, however it seems to be a bad predictive factor [36]. By contrast, *KRAS* mutation seems to be a good predictive factor for immune check-points inhibitors, reflecting a likely high tumor mutational board due to tobacco carcinogens, but some contradicting results are described in the literature [38,39]. 

## 2. Therapeutic Strategies and Clinical Results

Therapeutic targeting mutant *KRAS* has proven challenging due to its high affinity for nucleotide and the lack of tractable binding pockets for small molecules inhibitors [13]. Targeting KRAS could be applied directly or indirectly. Indirect approaches target critical steps of KRAS activation as inhibitors of the nucleotide-exchange cycle (SOS inhibitors, SHP2 inhibitors), inhibitors of RAS processing (farnesyltransferase inhibitors), or inhibitors of KRAS pathways (RAF inhibitors, MEK inhibitors, or ERK inhibitors) (Figure 2A). Emerging therapeutics are based on small interfering RNA therapies, autophagy, immunotherapy, adoptive cell therapy, or cancer vaccines [13]. We are focused on direct inhibition.

Direct inhibition is actually developed with switch-II mutant selective covalents inhibitors and pan-RAS inhibitors (Table 1). Targeting the wild-type KRAS protein creates unacceptable toxicity, as KRAS in essential in development. The best progress is obtained for mutant-specific *KRAS* G12C switch-II covalent inhibitors which would be expected to circumvent toxicity attributed to inhibition of all KRAS sub-types including wild-type KRAS [13]. Identification of a novel allosteric binding pocket (P2) behind switch-II in the KRAS-G12C mutated inactive GDP-bound state protein, allowed the development of compounds to irreversibly inhibit KRAS G12C [39,40,41]. Their enhanced potency is due to an enhanced interaction with the H95 residue of KRAS G12C with covalent binding and irreversible switch-off downstream signaling pathways. They bound KRAS G12C in the inactive GDP-state, blocked guanine nucleotide exchange and consequently blocked KRAS association with BRAF. The majority of KRAS G12C (75%) is GTP-bound in the steady state. Nevertheless, as described previously, KRAS-G12C has the highest level of intrinsic GTPase activity among KRAS mutations, and is so vulnerable to such covalent attack [42,43,44,45,46,47].

Multiple small molecules have been developed against KRAS G12C, such as ARS-1620, the first inhibitor, which has little clinical activity but remains an important translational research tool to study mechanism of resistance [42,43]. In early phase clinical trials, two potent molecules KRAS G12C inhibitors, AMG-510 (sotorasib) and MRTX849 (adagrasib), have shown promising results in NSCLC [11,12]. In vitro, they succeeded to covalently bind the acquired cysteine within the switch II and inhibit downstream MAPK signaling, as evidenced by diminished phosphorylation of ERK (p-ERK), S6RP (p-S6RP), and, in the case of sotorasib, MEK (p-MEK). Non-*KRAS* G12C mutations were insensitive to such inhibitors. In vivo, in murine models, both agents inhibited downstream MAPK effectors and shrank tumors [11,12,13,14,15,16,17,18,19]. Sotorasib did not affect PI3K signaling [19].

In preclinical analysis, AMG-510 (sotorasib) led to the regression of KRAS G12C tumors [11] and was the first molecule to enter clinical trials in patients with KRAS G12C tumors, particularly NSCLC patients. The phase I portion of the CodeBreaK100 trial revealed a favorable safety profile and established early evidence of anticancer activity, showing a greater objective response rate (ORR) in NSCLC (32%) than in colo-rectal cancer tumors (7.1%), a disease control rate (DCR) of 88.1%, and a progression-free survival (PFS) of 6.3 months [12]. Phase II, which enrolled NSCLC patients with previously treated KRAS G12C tumors, exhibited 37.1% ORR, with 3.2% complete response and 80.6% with disease control [10]. The median duration of response was 11.1 months and the median PFS was 6.8 months. Stable disease was the best response in 43.5% of patients. The median overall survival (OS) was 12.5 months (Table 1). AMG510 (sotorasib) had synergistic growth inhibitory effects with inhibitors of proteins that activate or are activated by KRAS, such as MEK, AKT, PI3K, SHP2, and members of the EGFR family. Remarkably, AMG510 (sotorasib) improved the anti-tumor efficacy of chemotherapy and targeted agents, resulting in a pro-inflammatory tumor microenvironment and producing durable cures alone as well as in combination with immunotherapy based on immune-checkpoint inhibitors [19].

MRTX849 (adagrasib) is another covalent selective irreversible inhibitor of KRAS G12C characterized by a long half-life that equals the 24 h synthesis rate of the KRAS protein. In vitro analysis shows that the loss of SHP2, MYC, or mTOR pathways genes further sensitized tumors to MRTX849 (adagrasib) [13]. An ongoing phase 1–2 study of MRTX849 (adagrasib) in KRAS G12C mutant cancers has shown an objective response rate of 45% in NSCLC patients in a KRYSTAL-1 trial, with a DCR of 96% [14]. Other KRAS-G12C covalent inhibitors were in development, such as JNJ-74699157 (Ars-3248) or LY3499446 but were discontinued due to the development of unexpected toxicities. Two novel inhibitors are in phase I, GDC-6036 (Roche) (NCT04449874) and D-1533 (InventisBio) (NCT04585035).

Although disease control is remarkable for AMG-510 (sorotasib) or MRTX849 (adagrasib), mechanisms of adaptive resistance are rapidly occurring.

## 3. Primary/Innate Resistance to KRAS G12C Inhibition

According to the pre-clinical experiments in isogenic cell lines, poor response as primary resistance might be explained by a rapid process of nonuniform adaptation whereby some cells escape inhibition by producing new KRAS G12C, which is promptly converted to the active, drug-refractory state, while others without sufficient expression of newly synthetized KRAS G12C are eliminated by the treatment [42]. The response rate to sotorasib in phase 2 CodeBreaK100 trial is lower than for other targeted treatments, such as tyrosine kinase inhibitors, perhaps due the molecular heterogeneity of KRAS-mutant tumors, which are often found in patients exposed to tobacco smoke (92.9% of the patients in this trial currently or formerly smoked) [10]. The genome damage that has been associated with tobacco carcinogens and that is commonly seen with *KRAS* G12C mutations may provide alternative pathways to drive tumor growth [43].

Exploratory analyses of some biomarkers of CodeBreaK100 trial were not statistically powered, and the 95% confidence intervals overlap across subgroups. The results should be interpreted with caution, but some data seem interesting. Objective response and tumor shrinkage under sotorasib in CodeBreaK100 trial were not different depending on the expression of PD-L1 (cut-off <1%). There was also no difference depending on the tumor mutational burden (cut-off <10 mutations per megabase) [10].

Differences of responses were observed for co-occurring mutations. In the exploratory biomarker analysis from CodeBreaK100 trial, a response was observed in 50% of the sub-group of patients with mutated *STK11* and wild-type *KEAP1* and in 39% of the overall population. Among patients with mutated *KEAP1*, a response was seen in 23% if combined with *STK11* mutation and in 14% if *KEAP1* combined with wild-type *STK11*. This finding is noteworthy because inactivating genomic alterations in *STK11* confer primary resistance to check-points inhibitors and docetaxel in NSCLC patients with *KRAS* mutations [10,27]. In another exploratory analysis from KRYSTAL-1 demonstrated that adagrasib has better efficacy in patients with co-mutation *KRAS* G12C and *STK11* than in *KRAS* G12C alone (ORR 64 versus 33%). No apparent trend toward a higher ORR was observed with *KEAP1* or *TP53*. Remarkably, tumors with *STK11* co-mutations exhibited increased T cells, suggesting that adagrasib may reverse STK11-mediated immune suppression [14]. Induction of epithelial-to-mesenchymal transition (EMT) in NSCLC KRAS G12C mutated NSCLC cell lines led to both intrinsic and acquired resistance to KRAS G12C inhibition [44].

The molecular heterogeneity may also predispose tumors to adapt quickly to the selective pressure of KRAS G12C inhibition.

## 4. Secondary Resistance to KRAS G12 Inhibition

Despite the clinical benefit observed for many patients treated with KRAS G12C inhibitors, acquired resistance to single-agent therapy occurred in most patients [11,19,42,45,46]. As the covalent inhibitors required KRAS-G12C to be in the inactive GDP-bound state, resistant mutations could arise in KRAS G12C that disable the GTPase activity or that promote the guanine exchange of GDP to GTP. Proposed resistance mechanisms to covalent KRAS-G12C covalent inhibitors have been identified in vitro through CRISPR screening and include the loss of either NF1 or one of the other RAS isoforms (NRAS and HRAS) [13]. Pre-clinical studies have nominated putative mechanisms of up-front resistance, including RAS-MAPK pathway reactivation [11,13,14].

Little is currently known about clinical resistance, but several papers were published in 2021. Numerous resistance alterations converge on the reactivation of RAS-MAPK signaling pathway, suggesting that this may be a central common mechanism of acquired resistance. The first study of clinical acquired resistance after treatment by KRAS G12C direct inhibition in cancer was described for adagrasib monotherapy in KRYSTAL-1 trail, with histologic and genomic analysis of samples (tissue or ctDNA) obtained at the time of disease progression and compared with available results of sequencing before adagrasib treatment for a small number of patients [47]. A deep mutational screening with a library of missense variants was used to define possible second-site mutations that confer resistance to KRAS G12C inhibition. In this study, 38 patients—27 patients with NSCLC, 10 with colo-rectal cancer, and 1 with appendiceal cancer—were included. Putative mechanisms of resistance were detected in 17 patients (17/38, 45%) and 10 NSCLC patients (10/27, 37%), with a least one mechanism of resistance. In addition, many of these patients (7/17, 41%) presented multiple coincident mechanisms, of whom two patients had NSCLC. NSCLC and colo-rectal cancer present some different resistance mechanisms, with more frequent simultaneous mechanisms in colo-rectal cancers, but the difference was not significant due to the small number of patients. It is possible that genomic instability or DNA-damage response mechanisms between colo-rectal cancer and NSCLC at baseline or in response to KRAS inhibition could explain these differences. It could also explain the more modest activity of KRAS G12C inhibition in colo-rectal cancers. Three main categories of mechanisms are described. First, secondary *KRAS* mutation or *KRAS* amplification; second, alternative oncogenic alterations activating the RAS signaling pathway but not directly KRAS itself; and third, histological transformation from adenocarcinoma to squamous cell carcinoma.

These resistance mechanisms could be classified in two groups: “on-target” groups with secondary mutation in KRAS altering inhibitor binding and “off-target” groups with activation of another isoform of KRAS (NRAS …) or other member of MAPK signaling pathway (Figure 3).

The “on-target" resistance mechanism groups involve other activating KRAS mutations on *trans*, potential loss of KRAS G12C through a mutational switch to a different KRAS mutation on *cis*, and new KRAS G12C production or amplification (Figure 3).

Resistance mechanisms independent of modulation of *KRAS* G12C mutation also occur. These “off target” mechanisms regroup genetic events that consolidate the upstream regulation of KRAS, such as activation of nucleotide exchange (SOS1, SHP2), loss of NF1 or PTEN, activation of mutations in other RAS GTPases, and bypass activation while the target remains inhibited, such as activation of downstream signaling pathways (via RAF, MEK, ERK, RB1, p21, and p27) or activation of parallel signaling pathways (Figure 3).

All mechanisms culminate in a stabilization and activation of transcription factors driving cell-cycle progression.

Adaptation to G12C KRAS inhibitors consist of modulation by RTK via two ways: stimulating SOS-mediated nucleotide exchange or through activation in a G12C-independant manner (activation of other wild-type KRAS or other isoforms of RAS, or PI3K, or other pathways) [20]. Other adaptation changes have been identified with single-cell modeling by the production of new KRAS G12C [42].

### 4.1. On-Target Mechanism

Acquired *KRAS* alterations, including mutations or amplification, are detected in half of the NSCLC patients (5/10, 50%) at resistance under adagrasib [47].

#### 4.1.1. Acquired Activating KRAS Mutations

Acquired codon 12 *KRAS* mutations prevent adagrasib binding. *KRAS* activating gain-of-function mutations are described in NSCLC *on trans* in a separate *KRAS* allele than G12C allele (G12V n = 1). *KRAS* G12A as *KRAS* G12W mutation was reported in one NSCLC patient with a mutation c.36T>G from cysteine to tryptophan (c.34_36 GGT-> TGT-> TGG) [47]. In a study testing cfDNA under adagrasib, *KRAS* G12F mutation is described in *cis* and KRAS G12V and KRAS G13D are described in *trans* in the same patient [48]. Nevertheless, the VAF of *KRAS* G12C mutation in cfDNA post treatment was much higher than those of the newly emerging alterations, pointing to KRAS G12C as a truncal mutation that is not extinguished by treatment and dominates with minor sub-clonal branches harboring the putative resistance alterations [49].

#### 4.1.2. Novel KRAS Mutations

Novel *KRAS* mutations are described in the switch-II adagrasib-binding pocket (R68S n = 1; H95D n = 1; Y96C, n = 1) [47]. The role of drug-binding site mutations seems different depending on the site of mutation and on the drug. Analysis of such mutations in the switch-II pocket by X-ray crystallographic structure of drugs revealed that all these mutations could result in disruption of noncovalent binding interaction of adagrasib and weaker interaction between H95 residue and sotorasib. Exogenously, expression of double-mutant *KRAS* allele G12C and new *KRAS* switch-II pocket mutations in Ba/F3 cell line showed that R68S, H95D, H95Q, and Y96C conferred marked resistance to adagrasib, as well as R68S and Y96C to sotorasib. In contrast, H95D/Q/R mutants remained sensitive to sotorasib. Biochemical analysis of response to adagrasib shows that R68, H95D/Q/R, and Y96C mutations block drug binding as indicated by the absence of KRAS band shift and complete prevention of drug-mediated suppression of RAS/MAPK signaling. In contrast, R68S and Y96C but not H95D/Q/R mutations mediated resistance to sotorasib [46]. Y96D residue, corresponding to a tyrosine-to-aspartate mutation at position 96, may represent a shared vulnerability to the different available KRAS G12C inhibitors, as demonstrated by structural modelling [48]. Based on crystal structure of different inactivate-state of KRAS inhibitors bound to KRAS G12C, the Y96D substitution appears to disrupt a critical hydrogen bound between the hydroxyl group of tyrosine 96 and the pyrimidine ring of adagrasib. More generally, the amino-acid change at the tyrosine 96 locus is thought to weaken drug–target chemical interactions by making the switch-II pocket of the mutant enzyme more hydrophilic. This modification also affects occupancy by other KRAS G12C inhibitors, representing a shared lability of several compounds. Consistent with a functional role of *KRAS* Y96D, ectopic introduction of the mutant gene into *KRAS* G12C-addicted cancer cell lines attenuated the growth-suppressing effect of inactive-state *KRAS* G12C inhibitors and enhanced KRAS signaling, indicating that *KRAS* Y96D is an oncogenic mutation that leads to constitutive KRAS activation and imparts resistance to KRAS G12C blockage [48,49]. Although Y96C appeared to be a clonal resistant event, the R68S and H95D/Q/R were identified in cfDNA at a low VAF [46,47] and occurred with other concurrent resistant mutations suggesting sub-clonally of these mutations. This diversity of KRAS mutations may pose a challenge in the development of effective next-generation inhibitors. These results suggested that differential drug-binding mechanisms between adagrasib and sotorasib can lead to the emergence of drug-specific resistance mutations.

Outside the drug-binding pocket, deep-mutational scanning also revealed *KRAS* resistance mutations. Even most of the second-site resistance KRAS mutation occurred at residues involved in drug binding; several mutations associated with resistance to adagrasib and sotorasib were detected in amino-acids located outside the drug-binding pockets in including known oncogenic mutations at codons 13, 59, 61, 117, and 146 that impede GTP hydrolysis or promote GDP to GTP nucleotide exchange [47]. In vitro analysis of BA/F3 cells expressing such mutations with a relative resistance score to G12R mutant show that Y96C drug-binding resistance mutation showed high-level resistance to adagrasib but not to sotorasib. Mutations at residues outside of the drug binding-pocket, and known to be involved in nucleotide exchange, including G13D, A59S, K117N, and A146P, conferred more modest resistance than the G12R and Y96C drug-binding mutations [46]. KRAS mutations G12V and G13D could also be detected at low VAF in cfDNA [47,48].

Finally, there is distinct mechanistic classes of *KRAS* resistance mutations as well as drug-specific patterns. Mutations within the drug binding pocket cause high-level of resistance. Other second-site mutations cause decreased GTP hydrolysis (G13D and Q61R) of enhanced GDP to GTP nucleotide exchange (G13D, A59S, and A146P). These last mutations probably increase the fraction of KRAS protein in the active GTP-bound form that does not bind to the drug. Nevertheless, some nucleotide exchange resistance mutations not yet observed in clinical samples, and whether they will cause resistance at clinically achievable concentrations of the drug remains to be elucidated.

#### 4.1.3. KRAS G12C Amplification and New KRAS G12C

High level amplification of the *KRAS* G12C allele was detected in one NSCLC patient with no other mechanism of resistance [47].

Another resistance mechanism could also be involved based on adaptive feedback activation of RTK ultimately activating MAPK signaling. As novel KRAS G12C inhibitors solely inhibit the inactive GDP-bound KRAS conformation, only cells with KRAS G12C inactivation state are strongly sensitive, stop proliferating, and enter into quiescence. Other cells adapt to the KRAS G12C inhibitor to reactivate KRAS transcriptional output. Thus, the population of tumor cells is in a non-uniform rate of inactive to active KRAS G12C cycling, and cells with KRAS G12C in the active conformation would be insensitive and could reactivate MAPK signaling pathway [42]. Cells express active GTP-bound KRAS G12C more abundantly, present low expression of p27 with two candidate genes that can mediate escape from KRAS G12C inhibition, heparin-binding EGF (HBEGF) and Aurora kinase (AURKA) [42]. An in vitro model shows that cells with low KRAS output produce new KRAS G12C protein—rather than the activation of other wild-type RAS isoforms not bound by the inhibitors as ARS 1620. Upstream signals, such as those mediated by EGFR and SHP2 or cell-cycle regulator aurora kinase A (AURKA), maintain the new protein in its active drug-insensitive state. The activation of the EGFR–SHP2 pathways maintains newly synthetized KRAS G12C protein in an active GTP-binding form. AURKA binds newly produced KRAS-G12C, which in turn stabilizes the interaction between CRAF and KRAS and subsequently mediates ERK effectors. In cells in which upstream signal are not active, the new KRAS G12C spends a longer time in its inactive conformation in which is can be bound by the inhibitor drugs. There is a non-uniform treatment response with diverging effects across the cancer population. These adaptive mechanisms of resistance seem to occur independently of co-occurring alteration in tumor suppressor genes *TP53* and *STK11.*

### 4.2. Off-Target

#### 4.2.1. Wild-Type KRAS and Activation of RAS Isoforms

The current generation of KRAS G12C inhibitors are selective covalent inhibitors of the G12C mutant and, as a result, do not inhibit wild-type RAS.

Wild-type and mutant KRAS subtypes coexist in the same cell, thus providing a feedback mechanism to reactivate RAS signaling if one of the two KRAS pathway is blocked. If KRAS-G12C is effectively and completely inhibited, residual wild-type RAS (NRAS and HRAS) activity may confer resistance to KRAS G12C inhibitors. As RTK activation is suppressed by ERK-mediated negative feedback, the MAPK signaling pathway could be moderately restored under KRAS G12C inhibitors by a release of upstream feedback inhibition of RTK which bypass KRAS G12C inhibition by activating wild-type KRAS (Figure 2 and Figure 3). In vitro analysis of adaptive response to KRAS-G12C inhibitors (ARS-1620 and AMG510) showed that rapid (4 h) downregulation of MAPK pathway activation (decreased expression of P-MEK, P-ERK, and P-ESK) is followed by MAPK pathway reactivation within 24–48 h, despite continuation of suppression of KRAS-GTP [45,50]. They demonstrated that such reactivation is due to increased NRAS-GTP and HRAS-GTP levels, suggesting that KRAS-G12C cells lines could adapt rapidly to selective inhibitions by activating wild-type RAS isoforms and that is sufficient to restore MAPK signaling. The increase activation of wild-type RAS activity is the result of activation of multiple different receptor tyrosine kinase (RTK) activation (EGFR, HER2, FGFR, and cMET). This specific RTK driving the rebound in MAPK signaling varied among the cells lines.

This feedback reactivation of wild-type RAS could occur in parallel to the neo-synthesis of KRAS-G12C protein. As SHP2 and SOS1 are the common nodes of RTK signals SHP2 inhibitors or SOS1 inhibitors may either enhance the activity of KRAS G12C inhibitors or reverse adaptive resistance. This hypothesis has been confirmed in pre-clinical models [51].

#### 4.2.2. MAPK Signaling Pathway: Up-Stream and Down-Stream De-Regulation

KRAS G12C inhibition can be overcome via feedback activation of either upstream or downstream mediators of the RTK-KRAS-MAPK cascade as already observed with targeted therapies for BRAF or EGFR mutations.

The first suggesting of such bypass signaling was described in vivo analysis demonstrated that ERK-dependent signaling is reactivated under adagrasib [11]. One hypothesis is that only cells with KRAS G12C in the inactive conformation would be strongly inhibited by the inhibitor, and no uniform rates of inactive/active KRAS G12C could exist in a tumor. Cells with KRAS in the active conformation would be insensitive and could mediate reactivation of MAPK signaling as shown in vitro analysis [41]. Cells exposed to ARS-1620 became initially quiescent, but a subpopulation subsequently escape this state. Single-cell analysis shows that cells with low-levels expression of p27 do not become quiescent, express active GTP-bound KRAS more abundantly, and are not eliminated by re-challenge with ARS-1620. Two candidate genes, heparin-binding EGF (HBEGF) and aurora kinase (AURKA), seem involved. These results suggest that EGFR signaling mediates adaptive resistance to KRAS G12C drugs. On another hand, AURKA accumulates in adapting cells, elicits accumulation of KRAS-GTP and P-ERK and lowers the potency of ARS-1620. This adaptive response to KRAS G12C inhibitors could be due to newly synthetized KRAS G12C that undergo immediate nucleotide change to an active GTP-bound conformation before being linked to RAS G12C inhibitors, with EGF being the likely driver of new RAS transcription and AURKA maintaining KRAS in the active GTP-bound conformation (cf supra). Indeed, origin of tissue predicts responsiveness with colo-rectal cancer cells showing rapid upregulation of P-MEK and P-ERK [46]. Colorectal cancer cells show increased basal phosphorylation of EGFR and respond to EGF by activating MAPK signaling, even in the presence of an activating *KRAS* G12C mutation that is not observed in NSCLC cells. A synergy between sotorasib and MEK inhibitors has been demonstrated in vivo [19].

Acquired bypass mechanisms of resistance were also detected in six NSCLC patients, including mutations of *RET* (M918T, n = 1), *MET* amplification (n = 2), *BRAF* (V600E, n = 1), *MAP2K1/MEK1* (E102_I103del, n = 1) and *PI3KCA* (H1074R, n = 1) [47]. A study related of KRAS isoform *NRAS* mutations (Q61K/R/L) in cfDNA, as well as MAPK signaling pathway with *BRAF* (V600E) and *MAP2K1* mutations (K57N, Q56P, and E101_I103 del) [48]. Given the very low prevalence (also cumulatively) of the identified mutations, their causal role in tumor progression and clinical relapse is not evident [47,48]. This result is based on cfDNA, and it may well be that the representation of mutant DNA was more prominent in the tumor cells than in the blood [49]. Loss of function mutation in PTEN (downstream) or NF1 (upstream) have also been described.

Upstream RTK regulators (EGFR, HER2, FGFR, cMET, and SHP2), direct mediators of KRAS activation (AURKA), and/or effectors of MAPK and PI3K pathways may escape form KRAS G12C inhibition, with different mechanisms depending on the tumor tissue type [9]. *cMET* amplification is described in vitro, leading to MET activation, reinforcing RAS cycling from its inactive form to tis active form. In addition to RAS mediated MEK-ERK induction, MET induced AKT activation independently of RAS [52].

Finally, a combination of KRAS-G12C inhibitors (adagrasib, ARS-1620) with EGFR targeted therapies (gefitinib, afatinib) was found to reduce downstream MAPK signaling (P-MEK and P-ERK) in vitro as well as in vivo, to reduce tumors in vivo in mouse xenografts, especially in colorectal cancer cells lines [11,42]. Co-administration in vivo of SHP-2 inhibitors with ARS-1620 was found to diminish adaptive reactivation of GTP-bound KRAS in xenografts, and also in vitro increase inactive GDP-bound KRAS, induce suppression of P-ERK and increasing T-Cell infiltration in NSCLC cell lines [41]. This effect augmented with a triple combination added EGFR inhibitors [42]. Combination of SHP2, PI3K, and KRAS G12C inhibitors was found to induce durable tumor regression in EMT-induced mouse xenografts, which exhibit FGFR and IGF1R-induced MAPK and PI3K reactivation [44]. Pan-KRAS inhibitors belong to another class of molecules inhibiting SOS1 from binding inactive GDP-bound KRAS [53]. Similarly, a combination of KRAS G12C inhibitors with either PI3K or mTOR inhibitors overcame the adaptive increase in PI3K signaling, and increased inhibition of MAPK/PI3K signaling [11].

#### 4.2.3. Fusion of Genes

Fusions appeared to be more common in colo-rectal cancers with *RET* (CCDC6-RET), *ALK* (EMl4-ALK) and multiple fusions (*RAF1*, *BRAF*, and *FGFR3*) than in NSCLC patients, but larger studies are necessary to confirm this tendency [47].

#### 4.2.4. Proliferative Signaling as Resistance Mechanism

In addition to adaptive reactivation of MAPK signaling, increased proliferation by disinhibition of the cell-cycle transition could be another mechanism of resistance, under KRAD G12C inhibitors. KRAS G12C inhibitors sequestrated tumor cells in a quiescent state (G0). In KRAS mutant NSCLC, up to 20% of patients presented loss-of-function mutations in the cell regulator CDKN2A gene, which leads to constitutive CDK4/6 RB phosphorylation. In vivo combination of adagrasib and palbociclib, a CDK4/6 inhibitor, showed synergy with P27 accumulation, decreased RB phosphorylation and decreased tumor volume in CDKN2A deficient models [11]. Other cell cycle inhibitors such as the AURKA inhibitor alisertib could be another approach [42].

#### 4.2.5. Phenotypic Transformation

Histologic transformation from adenocarcinoma to squamous cell carcinoma was described in one NSCLC patients without any identifiable genomic mechanisms of resistance. Tumor and ctDNA sequencing did not miss a major known molecular marker in such situations [47].

Epithelial to mesenchymal transition (EMT) is one of the acquired resistance mechanism to EGFR TKIs. The induction of EMT in sotorasib-sensitive cells by adding TGF-β or using transfection with SNAIL leads to acquired resistance to sotorasib through activation of PI3K, but not associated with AKT activation [44]. AKT seems not to be essential for such acquired resistance. The insulin-like growth factor receptor (IGF-1R) pathways mediates PI3K activation in a SHP2-independent manner in vitro, and leads to increased MAPK signaling via FGFR [44]. The mechanism of PI3K pathway mediated resistance to KRAS G12C inhibitors may depend on the tumor type and the degree of cell de-differentiation [51].

#### 4.2.6. Immune Mechanism of Resistance

KRAS G12C therapy resistance could impair antitumor immunity [9,51]. In vitro analysis shows that an impaired host immune system may confer resistance independent of MAPK reactivation or proliferative signaling [19]. Sotorasib appeared to induce infiltration of CD8+ T cells, macrophages, and dendritic cells, with genetic signature of IFN signaling, chemokines production, and antigen processing, suggesting that KRAS G12C inhibition could boost T-cell priming. Combining sotorasib with anti-PD1 therapy increased T-cell infiltration and led to complete and durable responses. Co-occurring mutations may modulate the immune response of tumors, as *KEAP1* and *STK11* mutations are correlated with cold immune microenvironment and TP53 with hot tumor infiltration [27].

In contrast to resistance to targeted therapies as first/second generation of EGFR tyrosine kinase inhibitors, diverse mechanisms are involved. In the majority of patients at resistance, at least one of these mechanisms did not involve the KRAS gene itself (14/17, 82%). Furthermore, numerous patients (7/17, 41%) had more than one mechanism of resistance, suggesting that KRAS G12C inhibition leads to strong selective pressure and convergent evolution of multiple distinct mechanisms of resistance [47]. Mechanisms of sotorasib are currently unknown and direct comparison between clinically observed adagrasin and sotorasib resistance mechanism is delicate. Paracrine growth factors secreted by the tiny portion of resistant cells protected the surrounding arrays of sensitive cells from the therapeutic. Finally, in the absence of ultra-deep sequencing data on the pre-treatment tumor samples, it remains unclear whether the mutant subpopulations preexisted at a very low frequency on the original tumor or exist de novo during treatment [47].

All studies confirm that KRAS G12C inhibitors can only delay tumor progression before the tumor evolves mechanisms to escape. Development of more potent and brain-penetrant KRAS G12C inhibitors could prolong tumor suppression and provide improved PFS, as well as the most efficacious combination partner with such inhibitors.

Origin of tissue can influence the mechanism of resistance, as fusions gene is present in colo-rectal cancers which are often micro-satellite instable, explaining the increased incidence of fusion genes. Colo-rectal cancers could develop resistance primarily via activation of upstream EGFR as well as NSCLLC deploying all mechanisms, depending on the presence of co-occurring mutations in *CDKN2A*, *STK11*, or *TP53*. Clinical trials must take into account tissue-specific differences in resistance mechanisms. As mechanisms of resistance are complex and diverse, combination of treatments in order to elicit long-term disease control are ongoing.

## 5. Perspectives

### 5.1. Treatments

Multiple resistance mechanisms are possible with KRAS G12C inhibitors, with *KRAS* mutations often sub clonal or occurring in the context of multiple putative mechanisms. The combination of several drugs interfering in different signaling pathways may prevent or delay the development of resistance but increase toxicity. The design of such trials should follow a rationale based on the genetic, metabolic, and immune mechanisms of resistance. One question is to know if combinatorial strategies will either be used at the time of acquired resistance or moved sooner to hopefully delay resistance.

Some resistant mutation in *KRAS* and *NRAS* mutations (with the exception of KRAS Y96D) are not actionable.

Co-targeting a single RTK such as EGFR would be unlikely to maximally suppress MAPK signaling pathway activity; co-inhibiting the phosphatase SHP2 could be useful to attenuate the activation of wild-type RAS with more durable ERK inhibition, as demonstrated in mice models with xenograft tumors [45]. Thus, upstream suppressed pathway activation is correlated with a minimal MAPK pathway activation [45]. While co-treatment with an MEK inhibitor can also prevent rebound in MAPK pathway signaling with low MAPK pathway activation, the resulting negative feedback inhibition of RTK and RAS can lead to the activation of parallel oncogenic pathways such as the PI3K/AKT signaling pathway [45,50]. However, MEK inhibitors by targeting a downstream node in the feedback pathway in the MAPK pathway can release upstream RTK and RAS from feedback inhibition which can, in some cellular context, result in parallel signaling pathways such as the PI3K kinase pathway. Therefore, co-targeting upstream may prove to be more effective than co-targeting downstream, as it would be predicted to not only induce more potent MAPK inhibition but also prevent reciprocal activation of parallel proto-oncogene signaling pathways. Combination therapies that target multiple nodes in the MAPK pathway through “vertical pathway targeting” with SHP2 inhibitors combined with KRAS G12C inhibitors could induce more profound responses in pre-clinical models [45]. Early phase clinical trials are ongoing to test KRAS G12C inhibitors with either EGFR inhibitors, SHP2 inhibitors, or SOS1 inhibitors. A clinical trial of sotorasib and MEK inhibitors is ongoing [19]. Early phase clinical trials of KRAS G12C inhibitors and mTOR inhibitors are ongoing. At this time, no clinical trial test AURKA inhibitors.

cMET inhibitor crizotinib can restore in vitro sensitivity to sotorasib in case of *cMET* amplification by eliminating RAS6MEK-ERK as well as AKT signaling [50].

As proliferative signaling could be a resistance mechanism, there is a significant translational potential for combining KRAS G12C inhibitors with either cytotoxic chemotherapy or inhibitors of interphase CDK [9]. Clinical trials are ongoing with combinations of KRAS G12C inhibitors and immunotherapy in NSCLC.

New approaches will be developed with vaccines or tumor-infiltrating lymphocytes therapy (TIL) (based on reintroducing patient intratumor T cells to the peripheral vasculature after boosting them with cytokines), targeted degradation of KRAS with ankyrin repeat proteins (DARPins) and C-12 direct covalent degrader molecules (PROTACs) [54].

Continued investigation into clinical mechanisms of resistance to KRAS G12C inhibitors in larger cohorts of NSCLC patients will be required to define the spectrum and the frequency of *KRAS* Y96X and other on-target mutations. It may be necessary to develop novel compounds that are able to target KRAS G12C/Y96D to overcome clinical resistance in clinics, such as RM-018 [1,2]. This drug has an affinity for the chaperon protein cyclophilin-A, and the resulting complex facilitates the formation of extensive protein–protein surface interactions that sterically occlude KRAS G12C in its active state and preclude KRAS association with downstream signaling pathways.

Most *KRAS* resistance mutations result in resistance to multiple KRAS G12C inhibitors, suggesting that there may be benefit of sequential treatment with such inhibitors. Nevertheless, possible treatment strategies have been proposed when acquired resistance is caused by secondary *KRAS* mutation as *KRAS* secondary mutations might offer differential sensitivity [1,2]. Adagrasib could be proposed after sotorasib in cases of *KRAS* G13D, A59S/T, or R68M mutations; sotorasib could be proposed after adagrasib in cases of *KRAS* Q99L mutation; and new drugs after adagrasib in cases of KRAS Y96D/S mutation.

### 5.2. Biology Testing

KRAS G12C inhibitors could afford direct quantification of drug-bound *KRAS* G12C in tumor biopsies. Other approaches, such as measuring the phosphorylation of ERK or other KRAS signaling intermediates, could also be analyzed. Noninvasive cfDNA assays offer an alternative approach to estimate the effect of treatment on *KRAS* G12C burden during time.

Low levels of VAF in cfDNA suggest putative polyclonal mechanisms of resistance with several minor mutated sub-clones, primarily concerning *KRAS* mutations but also notably affecting various nodes of the MAPK pathway and suggesting the use of rational combination of KRAS G12C inhibitors with downstream MAPK pathway inhibitors [48].

Clonal variations in the genetic composition of treated tumors may also modify the synthetic rate of newly produced KRAS G12C, and the ratio between active and inactive KRAS in functionally heterogeneous tumors subpopulations, which may influence adaptive fitness and susceptibility to inactive-state inhibitors.

Such analysis could distinguish tumors that grow independently of KRAS G12C and determine understanding of transient response in NSCLC patients.

## 6. Conclusions

Future investigations are expected to shed light on the various mechanisms of adaptation or resistance and the development of combination strategies to enhance the anticancer activity of KRAS G12C inhibitors [55]. The diversity of on-target and off-target mechanism support the need of development of additional KRAS inhibitors with alternative modes of binding and different allele specificities [56]. Development of effective combination therapy regimens will be required to fully combat resistance mechanisms that emerge during treatment with new direct KRAS inhibitors. It could be interesting to know if mechanisms of primary/acquired resistance is different among the drug used.

A phase 3 trial to compare sotorasib treatment with docetaxel in patients with previously treated, locally advanced, un-resectable or metastatic NSCLC with KRAS G12C mutation in under way (CodeBreaK 200, NCT 04303780). In addition, efforts are ongoing to investigate sotorasib in combination therapies (CodeBreak101, NCT 04185883). Identification of patients who may benefit from sotorasib regimens in the context of first-line treatment is also challenging.

As with other targeted therapies, preemptive strategies aimed at using inhibitors against the resistance onco-proteins as up-front therapies, before clinical manifestation of the corresponding mutations, should be considered. There is intense interest in understanding which combinatorial strategies will be either used at the time of acquired resistance, or moved sooner to hopefully delay resistance. For example, a combination that includes downstream inhibitors such as SHP2 might be effective.

Biomarkers for predicting which drug combination might be the most effective for individual patients will be important to define. KRAS-mutant NSCLC are characterized by high degrees of genomic heterogeneity, as represented by diversity of potential co-occurring mutations. Subclonal populations could be detected at time of resistance [48]. It will be important to determine the extent to which inter- and intra-tumoral heterogeneity influences the type and magnitude of adaptive drug response in order to maximize the efficacy of KRAS G12C inhibitors.

## Figures and Tables

**Figure 1 cancers-14-01321-f001:**
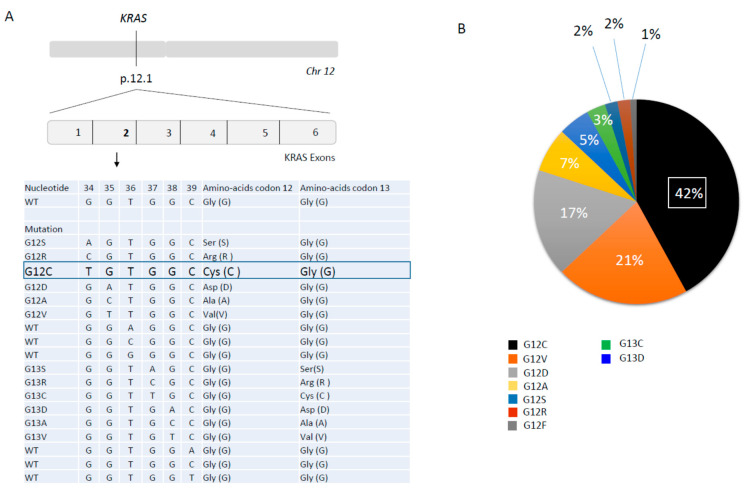
*KRAS* mutations at codon 12 and codon 13. (**A**) Nomenclature by nucleotide and by amino acids; (**B**) Repartition of the different *KRAS* mutations at codon 12 and codon 13 in NSCLC [17].

**Figure 2 cancers-14-01321-f002:**
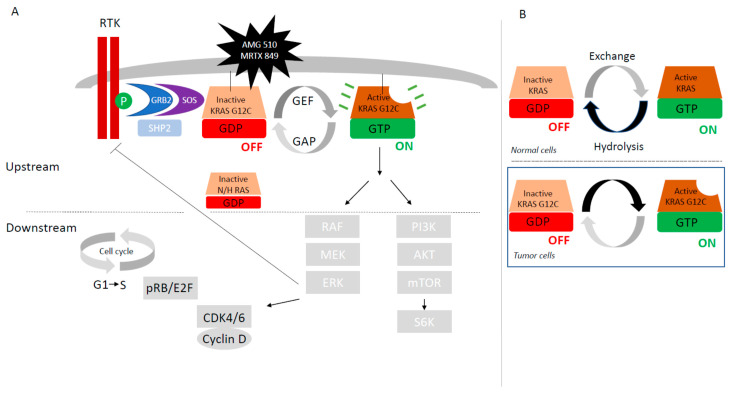
KRAS signaling. (**A**) KRAS signaling pathway. (**B**) KRAS “molecular switch”.

**Figure 3 cancers-14-01321-f003:**
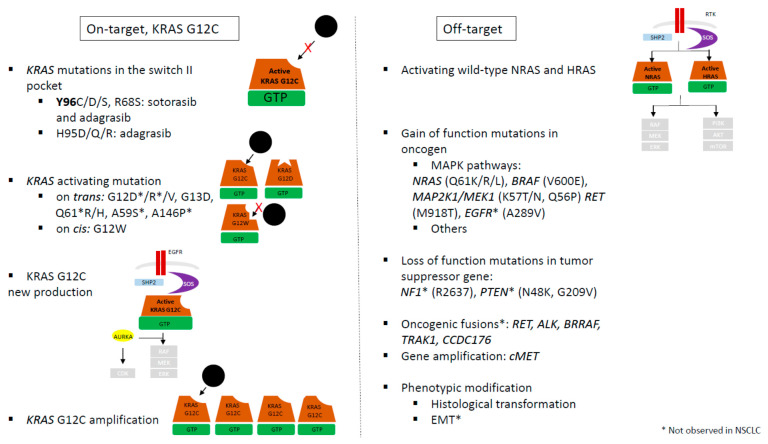
Mechanisms of resistance under KRAS G12C inhibitors.

**Table 1 cancers-14-01321-t001:** Activity of KRAS G12C inhibitors in early phase clinical trials; results of phases I/II with sotorasib or adagrasib [14,46].

KRAS G12C Inhibitors	AMG 510 (Sotorasib)	MRTX849 (Adagrasib)
Reference	[47]	[14]
Clinical trial	Phase II CodeBreaK 100	Phase I/II KRYSTAL-1
Patient population	KRAS G12C mutated advanced cancers	KRAS G12C mutated advanced cancers
Study population (n)	59 NSCLC	79 NSCLC, 51 evaluable
ORR (%)	37 (CR 3.2)	45
DCR (%)	80.6	96
mDOR (mo)	11.1	NR
mPFS (mo)	6.8	NR
mOS (mo)	12.5	NR

ORR: overall response rate; CR: complete response; DCR: disease-control rate; mDOR: median duration of response; mPFS: median progression-free survival; mOS: median overall survival; and NR: not reported.

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
