# Peer review of "Direct Targeting KRAS Mutation in Non-Small Cell Lung Cancer: Focus on Resistance"

_cancers, 2022, doi:10.3390/cancers14051321_

Round 1
Reviewer 1 Report
The submitted manuscript reviews the resistance mechanisms of KRASG12C inhibitors, sotorasib, and adagrasib, in either cell lines or human tissue samples of non-small cell lung cancer patients.
Overall, the manuscript is informative and comprehensive. While some issues need clarification, the sufficient information in this manuscript warrants publication.
Comments:
In section 1.6, please precisely explain the concepts of hot immune infiltration and cold immune infiltration.
In section 2, AMG-510 and sotorasib should be used appropriately. It may be confusing since these indicate the same inhibitor, i.e., its code name and generic name, respectively. This also applies to MRTX849 and adagrasib.
Please cite a reference for “sotorasib did not affect PI3K signaling.”
In the final sentence of section 2, the authors mentioned that, “Finally, molecules have been discovered that bind both the GDP-bound and GTP-bound state of KRAS, xxxx.” Please include the names of inhibitors that bind to the GTP-bound state of KRAS.
In section 4.2.6, the authors described that “All studies confirms that KRAS G12C inhibitors, similar to clinically approved fird-generation kinase inhibitors…” Please explain what are these third-generation kinase inhibitors.
In section 5.1, in the sentence, “New approaches will be developed with vaccines or TILs, targeted degradation ….,” what are TILs? Please explain precisely.
Author Response
Reviewer # 1
The authors thank Reviewer #1 for all the questions and their helpful comments. We answered the entire Reviewers’ comments and have revised our manuscript as suggested.
The reviewers' comments are as follows:
In section 1.6, please precisely explain the concepts of hot immune infiltration and cold immune infiltration.
As requested, we now precisely explain the concepts of hot immune infiltration and cold immune infiltration, in section 1.6.
In section 2, AMG-510 and sotorasib should be used appropriately. It may be confusing since these indicate the same inhibitor, i.e., its code name and generic name, respectively. This also applies to MRTX849 and adagrasib.
In section 2, terms of drugs are now always the same as code name and generic name for AMG-510 (sotorasib) and MRTX849 (adagrasib) respectively.
Please cite a reference for “sotorasib did not affect PI3K signaling.”
A reference is now cited for “sotorasib did not affect PI3K signaling”.
In the final sentence of section 2, the authors mentioned that, “Finally, molecules have been discovered that bind both the GDP-bound and GTP-bound state of KRAS, xxxx.” Please include the names of inhibitors that bind to the GTP-bound state of KRAS.
As these molecules are now not validated, we suppress this sentence.
In section 4.2.6, the authors described that “All studies confirms that KRAS G12C inhibitors, similar to clinically approved fird-generation kinase inhibitors…” Please explain what are these third-generation kinase inhibitors.
In the section 4.2.6, to more clarify the text, we have modified the first sentence of the paragraph with no mention to third-generation kinase inhibitors.
In section 5.1, in the sentence, “New approaches will be developed with vaccines or TILs, targeted degradation ….,” what are TILs? Please explain precisely.
In section 5.1. we now precise what TILs are “ tumor infiltrating lymphocytes therapy (TIL) (which based on reintroducing in peripheral vasculature, patient intra-tumor T cells after boost them with cytokines) “
Reviewer 2 Report
The manuscript of Reita et al. is interesting but not very original (in the last three years there are 82 reviews on "KRAS mutation in non-small cell lung cancer" in Pubmed).
The writing is poorly maintained. There are many spelling errors, words written in French, font formatting errors (bold, underline), repeated words (to to, the the), incorrect dash or space.
The bibliography must be improved, both in terms of quantity and formatting.
Sometimes the text is unclear and repetitive. I would strongly recommend improving for example the first part of the introduction and paragraph 1.2 because they are not clear. Also, I would put the paragraph 1.3 in 1.4 and 1.5 avoiding repetitions.
Moreover figure 2 needs improvement. In particular, the KRAS regulation (in Fig 2A) and the KRAS “molecular switch” difference between normal and cancer cells (in Fig 2B) are not clear. The inhibitors mentioned in paragraph 2 should be included in Fig 2A.
Figure 3 should be mentioned in paragraph 4.
Author Response
Reviewer #2
The authors thank Reviewer #2 for all the questions and their helpful comments. We answered the entire Reviewers’ comments and have revised our manuscript as suggested.
The reviewers' comments are as follows:
The writing is poorly maintained. There are many spelling errors, words written in French, font formatting errors (bold, underline), repeated words (to to, the the), incorrect dash or space.
All the manuscript is now corrected as for words and formatting errors, repeated words and incorrected dash or space.
The bibliography must be improved, both in terms of quantity and formatting.
The bibliography is now improved.
Sometimes the text is unclear and repetitive. I would strongly recommend improving for example the first part of the introduction and paragraph 1.2 because they are not clear. Also, I would put the paragraph 1.3 in 1.4 and 1.5 avoiding repetitions.
The first part of the introduction is improved as well as the paragraph 1.2. Paragraph 1.3 is now included in parapraph 1.4 in order to avoid repetitions.
Moreover figure 2 needs improvement. In particular, the KRAS regulation (in Fig 2A) and the KRAS “molecular switch” difference between normal and cancer cells (in Fig 2B) are not clear. The inhibitors mentioned in paragraph 2 should be included in Fig 2A.
The Figure 2A and 2B are now improved.
Figure 3 should be mentioned in paragraph 4.
Figure 3 is now mentioned if paragraph 4 section.
Reviewer 3 Report
Overall the authors summarize the latest findings on the role of KRAS mutations in the treatment of non-small cell lung cancer, which makes it an interesting topic. I have no substantial amendments to suggest; however, some minor corrections would be preferable.
Minor comments:
#1. In Figure 1B, I have difficulties in detect the difference among three panels. More detailed description is desirable.
#2. Regarding the combination of KRAS, STK11 and KEAP1 mutations, I suppose that the following manuscript needs to be cited and argued: Ricciuti B, Arbour KC, Lin JJ, et al. Diminished efficacy of programmed death-(ligand) 1 inhibition in STK11- and KEAP1-mutant lung adenocarcinoma is affected by KRAS mutation status. Journal of Thoracic Oncology, published on November 2, 2021 (online ahead of print). doi: 10.1016/j.jtho.2021.10.013
Author Response
Reviewer #3
The authors thank Reviewer #3 for all the questions and their helpful comments. We answered the entire Reviewers’ comments and have revised our manuscript as suggested.
The reviewers' comments are as follows:
#1. In Figure 1B, I have difficulties in detect the difference among three panels. More detailed description is desirable.
We precise the legend of figure 1B with focusing on KRAS G12C mutation.
#2. Regarding the combination of KRAS, STK11 and KEAP1 mutations, I suppose that the following manuscript needs to be cited and argued: Ricciuti B, Arbour KC, Lin JJ, et al. Diminished efficacy of programmed death-(ligand) 1 inhibition in STK11- and KEAP1-mutant lung adenocarcinoma is affected by KRAS mutation status. Journal of Thoracic Oncology, published on November 2, 2021 (online ahead of print). doi: 10.1016/j.jtho.2021.10.013.
We added the reference of Ricciuti B et al. [31] as requested for the combination of KRAS, STK11 and KEAP1 mutations.
Round 2
Reviewer 2 Report
The authors improved the text and figures.
There are still spelling mistakes and sentences to improve, for instance: lines 29, 279, 284, 330, 341, 373, 404, 504.
I would recommend removing "Figure 3" from the title of paragraph 4 because it is inserted in the text.
Some parts remain unclear (e.g. lines 51-53).
The bibliography has been increased by a single article.
Author Response
Reviewer # 2
The authors thank Reviewer #2 for all the questions and their helpful comments. We answered the entire Reviewers’ comments and have revised our manuscript as suggested.
The authors improved the text and figures.
There are still spelling mistakes and sentences to improve, for instance: lines 29, 279, 284, 330, 341, 373, 404, 504.
We now corrected all these spelling mistakes and sentences.
I would recommend removing "Figure 3" from the title of paragraph 4 because it is inserted in the text.
The term ‘Figure 3” is now removed from the title of paragraph 4.
Some parts remain unclear (e.g. lines 51-53).
These lines and some others are now corrected.
The bibliography has been increased by a single article.
We added another reference in the text and limited them to the date of the first submission of the paper.
- Zhao, Y.; Murciano-Goroff, Y.R.; Xue, J.Y.; Ang, A.; Lucas, J.; Mai, T.T.; Da Cruz Paula, A.F.; Saiki, A.Y.; Mohn, D.; Achanta, P.; et al. Diverse alterations associated with resistanceto KRAS(G12C) inhibition. Nature. 2021, 599, 679-683.